# Spatial Distribution of Immune Cells in Primary and Recurrent Glioblastoma: A Small Case Study

**DOI:** 10.3390/cancers15123256

**Published:** 2023-06-20

**Authors:** Delphine Loussouarn, Lisa Oliver, Celine Salaud, Edouard Samarut, Raphaël Bourgade, Christophe Béroud, Emilie Morenton, Dominique Heymann, Francois M. Vallette

**Affiliations:** 1INSERM UMR1307, CNRS UMR6075, Nantes Université, CRCI2NA, 44007 Nantes, France; delphine.loussouarn@chu-nantes.fr (D.L.); lisa.oliver@univ-nantes.fr (L.O.); celine.salaud@chu-nantes.fr (C.S.); edouard.samarut@chu-nantes.fr (E.S.); 2Centre Hospitalier Universitaire de Nantes, 44000 Nantes, France; raphael.bourgade@chu-nantes.fr; 3INSERM, MMG, Aix Marseille University, 13284 Marseille, France; christophe.beroud@univ-amu.fr; 4CNRS, US2B, UMR 6286, Biological Sciences and Biotechnologies Unit, Nantes Université, 44000 Nantes, France; emilie.ollivier@ico-unicancer.fr (E.M.); dominique.heyman@univ-nantes.fr (D.H.); 5Institut de Cancérologie de l’Ouest, 44800 Saint-Herblain, France

**Keywords:** glioblastoma, spatial profiling, immune markers

## Abstract

**Simple Summary:**

Glioblastoma (GBM) are the main primary brain tumors in adults. The role of the immune system in the progression of GBM is still largely under-evaluated. Here, we assessed the distribution of multiple immune biomarkers in a handful of coupled primary and recurrent GBM using both classical immunochemistry and Nanostring Geomix Digital spatial profiling. We observed an absence of correlation between the immune landscape during the evolution of GBM, both in the center and the periphery of the tumor. However, global analyses indicate differential localization of HLA-DR and B7-H3 between primary and recurrent GBM.

**Abstract:**

Only a minority of patients with glioblastoma (GBM) respond to immunotherapy, and always only partially. There is a lack of knowledge on immune distribution in GBM and in its tumor microenvironment (TME). To address the question, we used paired primary and recurrent tumors from GBM patients to study the composition and the evolution of the immune landscape upon treatment. We studied the expression of a handful of immune markers (CD3, CD8, CD68, PD-L1 and PD-1) in GBM tissues in 15 paired primary and recurrent GBM. In five selected patients, we used Nanostring Digital Spatial Profiling (DSP) to obtain simultaneous assessments of multiple biomarkers both within the tumor and the microenvironment in paired primary and recurrent GBM. Our results suggest that the evolution of the immune landscape between paired primary and recurrent GBM tumors is highly heterogeneous. However, our study identifies B3-H7 and HLA-DR as potential targets in primary and recurrent GBM. Spatial profiling of immune markers from matched primary and recurrent GBM shows a nonlinear complex evolution during the progression of cancer. Nonetheless, our study demonstrated a global increase in macrophages, and revealed differential localization of some markers, such as B7-H3 and HLA-DR, between GBM and its TME.

## 1. Introduction

Glioblastoma (GBM) is the deadliest and most common primary brain tumor in adults, with a poor median overall survival (OS) of approximately 13–15 months. GBM invariably recurs, despite extensive surgical resection and aggressive treatments such as radiotherapy and temozolomide (TMZ) therapy in combination (or not) with anti-angiogenic therapy (bevacizumab) or immunotherapy [1]. There is no standardized second-line therapy for recurrent tumors, and no significant results have been achieved, despite numerous investigations [2]. Surgical resection of recurrent GBM can be helpful, but is not always possible [3]. Immunotherapies have been evaluated in GBM in many studies, and the clinical results have been disappointing both for primary and recurrent GBM [4,5]. Nonetheless, despite the dismal results globally, it has become evident that some patients could benefit from immunotherapies [4,5]. These results emphasize the need to better characterize the immune populations in GBM patients.

In central nervous system cancers, immune surveillance has been overlooked for many years because of the “brain immune privilege” theory [6]. However, recent results have established that several pathological situations in the brain are actively controlled by the local immune response [7], and that brain tumors are potently immunosuppressive through local and distant mechanisms [8,9,10]. This is one of the primary reasons that immunotherapy protocols in GBM have shown little efficacy [11,12]. However, the quantitative and qualitative natures of the immune local population in brain tumors are still sparse [10]. The major non-neoplastic cell populations in the GBM microenvironment are tumor-associated macrophages (TAMs), which can have pro- or anti-tumoral functions depending on the GBM context [13]. Studies that have investigated the subpopulations of the immune infiltrates using different methodologies, including whole genome sequencing, transcriptomic, and immunohistochemistry (IHC), have established a somewhat contrasting immune landscape in gliomas, with some features shared with other brain pathologies due to common inflammatory processes involving macrophages and microglia [14]. In GBM, TAMs (both circulating macrophages and resident brain microglia) represent nearly half of the tumor mass [15], and the presence of CD68, a microglial marker, using a combination of tissue microarrays (TMA) and transcriptomic analyses, has been correlated with the clinical outcome of patients [16].

However, few studies have investigated the evolution and intra-tumor localizations of the different immune populations in GBM during treatment. To address this important question, we used a tissue bank composed of paired de novo (primary) and subsequent relapse (recurrent) tumors from GBM patients.

Recently, several techniques have been developed to allow multiplex detection for the visualization of different immune cells within tumors. The phenotypic profiling of the immune populations within the tumor is likely to provide important information, not only for tumor sub-classification but also for treatment decisions and clinical outcome predictions [17]. Studying multiple protein distribution in a single tissue and in well-defined regions can be partially achieved through multiplex IHC. NanoString Technologies developed a new technology, GeoMxTM digital spatial profiling (DSP), which enables the simultaneous and guided detection of up to 40 different proteins in different regions of interest in a single formalin-fixed, paraffin-embedded (FFPE) tissue [17].

We analyzed the immune infiltration from the center to the periphery of the tumors by delimitating selected regions of interest in primary and recurrent GBM. This work sheds light on the various immune components in the GBM tumor microenvironment, their evolution, and their possible roles in tumoral development.

## 2. Materials and Methods

### 2.1. Patient Population

Patients eligible for inclusion corresponded to the following criteria: patients were treated twice. First, for the primary tumor (surgery for intracranial glioblastoma with anatomopathological confirmation at the Nantes University Hospital) and second, for a recurrence after treatment consisting of radiotherapy plus chemotherapy (Temozolomide, TMZ), according to the Stupp protocol, at the “Institut de Cancérologie de l’Ouest” (ICO)-René Gauducheau. The recurrent tumor (second surgery with anatomopathological confirmation at the Nantes University Hospital) was treated between October 2013 and April 2019 (Stupp et al., 2015) [18]. The local ethics committee (GNEDS: Groupe Nantais d’Ethique dans le Domaine de la Santé) approved the collection and use of these data (approval on Dossier 06/15 on 8 April 2015). Written informed consent was obtained from all patients included in this study.

### 2.2. Tissue Microarray (TMA)

For TMA construction, selected regions were punched from the donor paraffin blocks with a coring needle of 2 mm diameter, and the material was then transferred into the recipient paraffin blocks. For each tumor sample, a pathologist selected four regions of interest (ROIs) corresponding to the core (tumor cells and microvascular proliferation) and the peritumoral area.

### 2.3. Immunohistochemistry and Its Quantification

Immunohistochemical analysis was performed on whole slide tissue sections of 15 pairs of matching primary and recurrent GBM. Sections (4 µm) were cut from the FFPE blocks and placed on Superfrost slides. The IHC technique was performed using an automated Dako Omnis automated system (Agilent Technologies, Santa Clara, CA, USA), using the streptavidin-biotin amplification technique after appropriate antigen retrieval. It involved the application of the following primary antibodies against CD3, CD8, CD68, PD-1 and PD-L1. The peroxidase activity was revealed using 3,30-diaminobenzidine for 5 min. The slides were then counterstained with hematoxylin and cover-slipped. The slides were scanned with NanoZoomer Hamamatsu 2.0 HT (Massy, France), processed using NDP.view2 software, and then quantified using QuPath v0.3.2 software.

### 2.4. Antibodies Labelling with Digital Space Profiling GeoMX^TM^

Immune profiling was performed at the Nanostring facility (Seattle, WA, USA) using antibodies raised against the antigens described in Appendix A. The selected compartments were chosen for high-resolution multiplex profiling, and oligos from the selected region were released upon exposure to UV light. For spatial transcriptomic analyses, 5 patients with available FFPE specimens from primary and paired recurrent GBM were selected. The protocol followed is described in https://nanostring.com/wp-content/uploads/WP_GeoMx_Antibody_Validation_White_Paper.pdf (accessed on 30 June 2020). First, FFPE tissue sections mounted on positively charged slides were processed using the Ventana Discovery Ultra platform (Ventana, Roche Diagnostics, Bale, Switzerland) before deparaffinization. Second, tissues were washed and incubated with 2 µg/mL proteinase K in phosphate-buffered saline before fixation and incubation overnight at 37 °C with GeoMx Immune antibodies (Appendix A). A minimum of eleven regions of interest (ROIs) were selected across the whole slide in stroma/tumor regions. After ultraviolet illumination, barcodes were collected in 96-well plates and dried for 1 h at 65 °C. Photocleaved oligos were resuspended in 10 µL nuclease-free water and hybridized to GeoMX Hyb codes at 65 °C for 18 h, before processing on the nCounter MAX system (NanoString). nCounter counts were converted to digital count conversion files using the NanoString GeoMx NGS pipeline (v2.1). After ROI quality control according to the recommendations of NanoString and a principal component analysis to eliminate potential outliers, areas of illumination (AOIs) raw counts were normalized using full quantile normalization (Limma R package v3.46). Heatmaps were generated using the pheatmap R package (v1.0.12) after averaging the gene expression of the indicated AOIs per patient.

### 2.5. Nanostring Genomic Arrays

RNA was isolated from FFPE tumor sections by dewaxing using deparaffinization solution (Qiagen, Valencia, CA, USA), and total RNA was extracted using the RecoverALL™ total nucleic acid isolation kit (Ambion, Austin, TX, USA).

Photocleaved oligonucleotides were then collected via microcapillary tube inspiration using an early version of the DSP platform (NanoString, Seattle, WA, USA) robotic system, then transferred into a microwell plate with a spatial resolution of approximately 10 μm. Photocleaved oligonucleotides from the spatially resolved compartments in the microplate were then hybridized to 4-color, 6-spot optical barcodes using the nCounter^®^ platform, enabling up to 800 distinctly label counts per compartment of the protein targets, representing the antibodies to which the tags were originally conjugated. Digital counts from barcodes corresponding to protein probes were first normalized with internal spike-in controls (ERCCs) to account for system variation, and then normalized to the area of their compartment.

## 3. Results

### 3.1. Patient Characteristics

Fifteen patients were included in this study. Their main characteristics are detailed in Table 1. There were ten men (67%) and five women (33%), all with clinical symptoms caused by the tumor (seizure or neurological deficits). The mean age at diagnosis was 54.93 years. Localizations of the primary tumors were as follows: five in the right frontal lobe (33%), four (27%) in the left frontal lobe, three (20%) (in the left temporal lobe, one (7%) in the right parietal lobe, one (7%) in the left parietal lobe, and one (7%) in the right parietal and occipital lobes. Localizations of the recurrent tumors were systematically in the edges of the proencephalic cavity, except for one patient (patient 10), in whom it in the same lobe, but far from the edges. The mean delay between both surgeries was 13.9 months (±4.8). The mean survival delay was 10.9 months (±5.25) after the second surgery. The mean overall survival was 24.42 months (±8.72).

All patients expressed wild-type IDH both in primary and recurrent tumors. In the primary tumors, ATRX expression was maintained in 14 patients (93.34%) and non-determined in 1 patient (6.66%). This was still observed in the recurrent tumors (except in patient 3 and 11, in whom partial loss was observed). Overexpression of p53 in primary tumors was observed in 12 patients and absent in 3 patients. At recurrence, 7 tumors did not express p53 and 2 tumors overexpressed p53.

Altogether, on these criteria, our patients were representative of the GBM population.

### 3.2. Increased Percentage of CD8^+^ and CD68^+^ Cells in Recurrent versus Primary Tumors

We analyzed the expression of T lymphocyte markers CD3 and CD8 with that of CD68, a macrophage/microglia marker, in a series of 15 pairs of matching primary and recurrent GBM. All patients completed the Stupp protocol for the management of GBM [18], which is based on optimal surgical excision associated with concomitant treatment with radiotherapy and alkylating chemotherapy (TMZ), followed by adjuvant chemotherapy between the first (primary) and second surgery (recurrent).

The percentage of the different antigens was determined in the tumor samples using IHC analyses on FFPE sections from the primary and recurrent tumors, as described in Methods section. Images from the IHC analyses of two tumors are presented in Figure 1A,B for CD3, CD8 and CD68. We observed a global increase in the percentage of CD68^+^ cells between primary and recurrent tumors, while the proportion of CD3^+^ and CD8^+^ cells was not significantly affected (Figure 1C). For quantification of the expression of the different markers, only the whole tumor was used. Note that almost all tumors (both primary and recurrent) showed a high percentage of CD68^+^ cells (Figure 1C). Of note, tumors that had a high CD8 expression also had a high CD3 expression (Figure 1D). Analysis of the expression of PD-1 and PD-L1 indicated that the percentage of labelled cells was always low (Figure 1D). However, in the majority of patients, an increased percentage of PD-L1^+^ cells was observed in recurrent tumors compared to the paired primary tumor (Figure 1D).

Our results show an increase in CD68^+^ cells but no increase in CD3^+^ or CD8^+^ cells between primary and recurrent GBM. The proportion of PD-L1^+^ cell population was augmented in recurrent vs. primary paired tumors (Figure 1D). Altogether, our results confirm previous results that CD3^+^, CD8^+^ and CD68^+^ infiltrating immune cells are present, but not in large amounts, in primary tumors and that CD68^+^ cells (i.e., macrophages) increase during recurrence. However, this global analysis did not account for the specific nature of the immune population in the different tumors, in the different regions from the core to the invasive front and/or the microenvironment.

### 3.3. Transcriptional Profiles of Recurrent vs. Primary Glioblastoma

To answer the question of precise localization of immune cells in the tumor, we selected five paired primary and recurrent tumors from the 15 patients listed in Table 1. First, we performed a transcriptomic analysis using the nCounter PanCancer Pathways Panel, which allowed the analyses of 770 genes from 13 cancer-associated canonical pathways (https://dev.nanostring.com/products/ncounter-assays-panels/oncology/ncounter-pancancer-pathways-panel/ (accessed on 30 June 2020)). As shown in Figure 2A, unsupervised hierarchical clustering analyses indicate that the recurrent tumors^®^ exhibit similar expression patterns in patients 1, 2, 4 and 5; these are different from the primary GBM, which were similar for patients 1, 2, 3 and 4 (Figure 2A). In contrast, the primary and recurrent tumors of patients 1 and 5 appear similar and are thus not affected by the treatment, at least for the genes present in the panel. The gene expression of the 770 genes encompasses all major cancer pathways, including Wnt, Hedgehog, apoptosis, cell cycle, RAS, PI3K, STAT, MAPK, Notch, TGF-β, chromatin modification, transcriptional regulation, and DNA damage control. However, when these 13 cancer-associated canonical pathways were analyzed separately, patients 4 and 5 exhibited closely related expressions between primary and recurrent tumors, while only one tumor (patient 2) appeared to undergo a significant difference during tumor progression (Figure 2B). Detailed analyses of the TGF-β and DNA damage pathways were performed, and again the primary tumor and recurrent tumor from patient 2 appeared to be different (Figure 2C,D). Analysis of TGF-β-related gene expression indicated a significant difference during GBM progression, while no correlation between primary and recurrent GBM was found for DNA damage (Figure 2C). Analysis of drivers and chromatin genes again indicated a difference between the primary and recurrent tumors in patient 2, when compared to the other tumors, which maintained similar gene expression between primary and recurrent GBM (Figure 2D–F). Thus, 4 of the 5 tumors exhibited common features during progression. The detailed analyses did not indicate specific traits between primary and recurrent status, apart from patient 2.

### 3.4. Immune and Cancer Spatial Distribution in Intra- and Peri-Tumoral Regions in Paired Primary and Recurrent GBM

Next, we used NanoString GeoMx™ DSP technology (https://nanostring.com/products/geomx-digital-spatial-profiler/geomx-dsp (accessed on 30 June 2020)) to further study the spatial distribution of immune markers in five paired primary and recurrent tumors. NanoString digital spatial profiling technology uses oligonucleotides tags bound to specific antibodies, which allows for the simultaneous analysis of 39 antibodies on FFPE tumor sections (Figure 3). We selected five pairs representative of primary and recurrent GBM (Table 1). For the five patients, we selected 12 regions of interest (ROIs) in sections from primary and recurrent tumors, representing regions located from the core to the periphery of the tumor, based on the following requirements:Homogenous cellular populations;Distinct tumoral or normal features;Intra- to peri-tumoral or microenvironmental localizations;Vascularized regions.

Figure 3A shows the selected ROIs for each patient in both primary and recurrent tumors. Analyses of the ROIs indicated an important variability in the expression of antigens, with few ROIs enriched for immune markers (Figure 3B). Selections were performed for the five tumors, as illustrated for patient 1 in Figure 3C, for 12 ROIs in the tumoral zone or peritumoral zone for primary and recurrent paired tumors. The ROIs analyzed for the 5 tumors can be find in Appendix A. Quantification of the antigens was performed on the five patients in 12 different tumoral or peritumoral ROIs, using the QuPath v0.3.2 software. The distribution of key immune markers in the different ROIs from the center of the tumor to the periphery, corresponding to macroscopically normal tissue in primary and recurrent tumors from patient 1, are represented in Figure 3C. The distribution of key markers (immune or cancer) appeared to be extremely variable among the patients examined (Figure 3D). In the primary tumors, patient 5 appears to be “colder” than the other patients. The presence of immune markers was found mainly in the peritumoral ROIs for patient 4, while the inverse was observed for patients 1, 2 and 3. In the recurrent GBM, more immune markers were found in patient 5, but mostly in the periphery of the tumor. The other patients exhibited very few immune markers at the periphery and in the center of the tumors, with the exception of patient 1 (Figure 3D). Thus, immune markers appeared to be diversely localized in the five tumors, and no obvious patterns could be found in the evolution of immune infiltrates from paired primary to recurrent GBM.

Next, we examined the co-expression of selected markers in the five patients (Figure 4). First, we analyzed, using matrix correlation, the co-expression of a selection of key immune antigens in primary and recurrent paired tumors, independently of ROIs. This included the expression of B7-H3, a co-stimulator molecule of the cell surface B7 protein superfamily, which is highly expressed in GBM [19]; CD68, a macrophage marker; CD11c, a pro-tumoral macrophage marker [20]; and T-lymphocyte markers such as CD3, CD8 and CD4. As shown in Figure 4A, B7-H3 appeared to be associated with the T-lymphocyte CD8+ /CD11c+ and macrophages CD68+/CD11c in primary but not in recurrent tumors. Likewise, the presence of CD56 was associated with PD-1 and PD-L1 in primary but not in recurrent tumors. Macrophage markers such as CD68/CD11c appeared to exhibit non-identical correlation patterns in both primary and recurrent paired tumors. In particular, the association of CD68 with PDL-1 in recurrent GBM indicates stronger immunosuppressive functions than in primary GBM. The lymphocyte markers CD3/CD8 were mostly co-expressed in both primary and recurrent tumors, with little overlap with CD4 expression (Figure 4A). HLA-DR, which is typically expressed by antigen-presenting cells, has been associated with M1-like macrophages and T cell activation. HLA-DR is associated with CD68 in primary GBM, and with CD4 in recurrent GBM, suggesting a switch between activation of M1-like macrophages and T-cell activation during tumor progression (Figure 4A).

Next, we examined the expression of key markers in each different ROI, patient by patient, and by ROIs (Figure 4B). The expression of STING (stimulator of interferon genes), a key component in DNA-mediated innate immunity [21], and B7-H3 was found in the same ROIs. In some cases, the co-expression of B7-H3 and VISTA, another immune checkpoint protein and a marker of poor prognosis in GBM [22], was also observed (Figure 4B). Macrophage/microglial markers including CD68, CD11c and HLA-DR were found to be the predominant immune cell markers in almost all the ROIs, and to be mainly co-localized (Figure 4B). Of note, the expression of these markers was not always correlated with that of cancer markers such as AKT and β-catenin, ruling out a non-specific labeling due to the presence of cancer cells (Figure 4B).

Altogether, our results confirm that the expression of immune markers was heterogeneous in the five patients examined, and that no strong correlation could be found to differentiate between the center versus the periphery of the tumors, nor between primary versus recurrent paired tumors.

Of note, patient 2, who exhibited distinct genomic features (Figure 2), did not show any particular immunological specificity.

### 3.5. The Expression of B7-H3 and HLA-DR in Primary versus Recurrent Paired Tumors

Next, we quantify the expression of key immune-related proteins, using immunofluorescence, in each ROI for patients 1–5 from the center to the periphery of the tumors. As shown in Figure 5A, B7-H3 and STING are the most expressed antigens in primary tumors (both in the center and the peripheral zones), while CD68 and CD11c were found only at the periphery of the tumor. In recurrent GBM, the predominant markers were HLA-DR and CD68, present both in the center and in the periphery of the tumors, while CD11c was found only in the center of the tumor (Figure 5A,B). It is noteworthy that CD11c localization appears to be different in primary versus GBM, as it is predominantly present at the periphery of the primary tumors, but in the center of recurrent tumors.

To confirm our observations, we analyzed the expression of B7-H3 and HLA-DR immunohistochemistry in the 15 paired primary/recurrent GBM listed in Table 1. As shown in Figure 5C and Appendix A, we found an increase in the percentage of HLA-DR^+^ cells between recurrent and primary GBM, while the percentage of B7-H3 positive cells was not modified. Indeed, no statistical difference was observed in the overall expression between paired primary and recurrent tumors.

## 4. Discussion

The role of cancer immune surveillance processes as a major safeguard from the early stages of malignancy to the final recurrent and metastasis stages has long been acknowledged [23]. However, very often, local and systemic immune systems can be overpowered by tumors, and immuno-editing can turn immune cells from foes to allies of cancers [24]. The purpose of immunotherapy is to use or enhance immune surveillance processes, or to bypass cancer induced-immunosuppression mechanisms to inhibit tumor progression [25]. In the past few years, astonishing progress has been made in many cancers by using immunotherapeutic strategies. Unfortunately, in too many cases, the response fell short because of the emergence of resistance in the targeted tumors [26,27]. GBM is the most common type of malignant brain tumor, and has a very poor prognosis. Most patients relapse within 12 months despite aggressive treatment, and patient outcomes after recurrence are significantly worse. The relationship between the immune system and GBM has been investigated for many years, and this tumor seems to be a model for resistance to immunotherapy [11]. However, partial responses observed during clinical trials suggest that the different components of the immune microenvironment should be taken into consideration to enhance immunotherapeutic efficiency [28]. The study described here aimed to gain a better understanding of GBM development and progression, and particularly of the nature of immune infiltrates within five paired primary and recurrent GBM.

Our findings suggest that immune markers are diversely expressed in GBM, with an increase in CD68^+^ cells indicating that macrophages/microglia increased during tumor recurrence. TAMs within the CNS originate either from resident microglia or from blood-derived monocytes, which are transformed within the tumor microenvironment to monocyte-derived macrophages [29].

NanoString GeoMx™ DSP technology has allowed us to refine the localization of immune cells within or around the tumors. We observed that spatial immune profiling again indicated an extreme heterogeneity, but revealed that the immune check point B7-H3 is present in GBM in primary or recurrent tumors, and in tumoral or peri-tumoral compartments (Figure 5).

HLA-DR is the marker most widely used to describe activated microglia in human brains [30], and B7-H3 is a promising new target in GBM [19]. The expression of HLA-DR and B7-H3 has been associated with a bad prognosis in GBM [19,31].

In non-malignant tissues, B7-H3 has a predominantly inhibitory role in adaptive immunity, suppressing T-cell activation and proliferation. In malignant tissues, B7-H3 inhibits tumor antigen-specific immune responses, leading to a pro-tumorigenic effect. B7-H3 also has non-immunologic pro-tumorigenic functions, such as promoting migration and invasion, angiogenesis, chemoresistance and endothelial-to-mesenchymal transition, as well as affecting tumor cell metabolism.

As shown in Figure 5C, the expression of B7-H3 was significantly increased in recurrent tumors, which has been associated with tumor aggressiveness and poor prognosis [32], while that of HLA-DR was not different between primary to recurrent GBM. Interestingly, HLA-DR expression correlates with macrophage markers in primary GBM, and only with CD4 in recurrent GBM, while B7-H3 strongly correlated with macrophage markers only in primary tumors (Figure 4). As a result, distinct correlations and intra-tumoral localization suggest that B7-H3 and HLA-DR are implicated in different types of immune reaction in primary and recurrent GBM. Nonetheless, the increase in HLA-DR in recurrent GBM suggests that HLA-DR could be a potential immune therapy target in IDH wild-type GBM patients. The exact roles of B7-H3 and HLA-DR in tumor progression are still uncertain. Of note, both are implicated in the activation of monocytes and macrophages, and the switch from the M1 phenotype to the M2 phenotype. Thus, our results agree with the fact that tumor-associated macrophages are essential for GBM progression. Hence, in this context, both HLA-DR and B7-H3 could be attractive targets in primary and recurrent tumors [30,33].

## 5. Conclusions

We have studied immune markers in primary and recurrent GBM patients and found that the distribution of immune markers in these tumors is extremely heterogeneous, which could explain why GBM are refractory to universal immuno-therapies. Nonetheless, our analyses revealed two putative targets in primary and recurrent GBM: HLA-DR and B7-H3. We believe that spatial profiling is providing new and highly informative results on complex tumor structures and compositions.

## Figures and Tables

**Figure 1 cancers-15-03256-f001:**
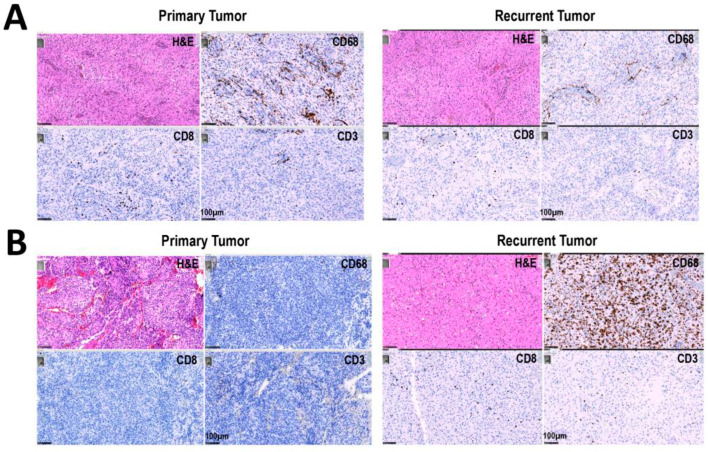
Immunohistochemical analysis of primary and recurrent paired tumors. (**A**,**B**) Immunohistochemistry analyses for CD3, CD8 and CD68 positive cells, as described in materials and methods, in two representative primary and recurrent GBM; Hematoxylin–Eosin (H&E) staining. (**A**) Primary: first surgery at diagnosis, recurrent: (**B**) second surgery after the first line of treatment. (**C**) Percentage of CD3^+^, CD8^+^ and CD68^+^ cells in primary (circle) and recurrent (box) GBM. CD68 staining is significantly increased in recurrent tumors compared to primary tumors. (**D**) Proportion of PD–1, PDL–1, CD3, CD8 and CD68 in each primary and recurrent GBM. ** *p* < 0.01; ns = not significant.

**Figure 2 cancers-15-03256-f002:**
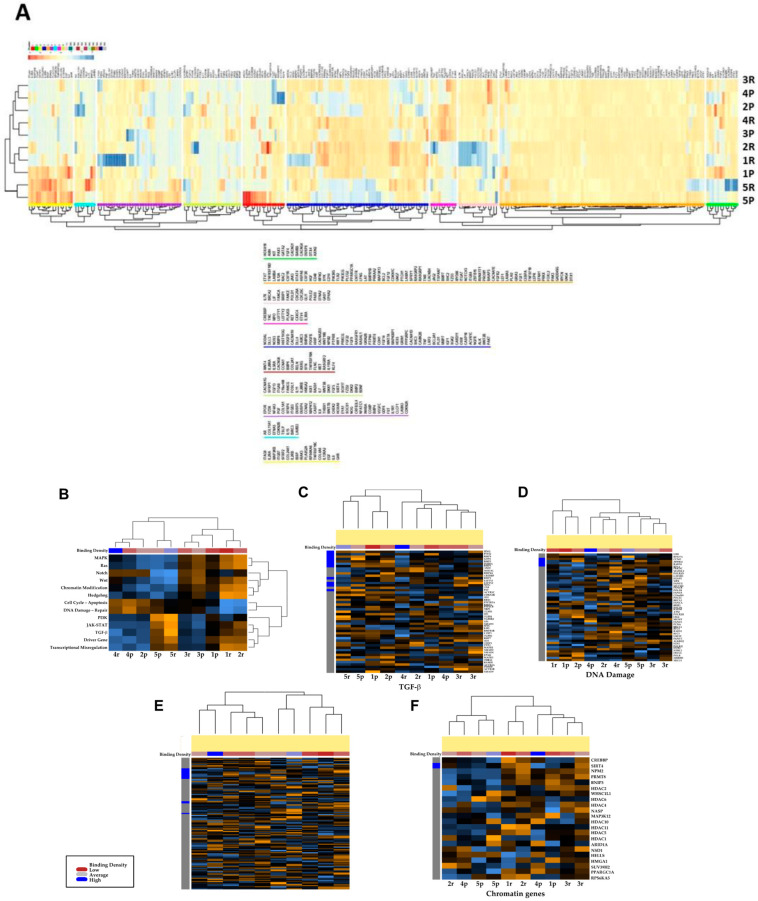
Transcriptomic analysis of primary and recurrent paired tumors in five selected patients using Nanostring nCounter^®^ PanCancer Pathways Panel. (**A**) Unsupervised hierarchical clustering of the five paired primary vs. recurrent GBMs. The heatmap was made with the 732 probe sets, (**B**) canonical pathways, (**C**) TGF-β pathway, (**D**) DNA damage, (**E**) Driver genes and (**F**) Chromatin genes.

**Figure 3 cancers-15-03256-f003:**
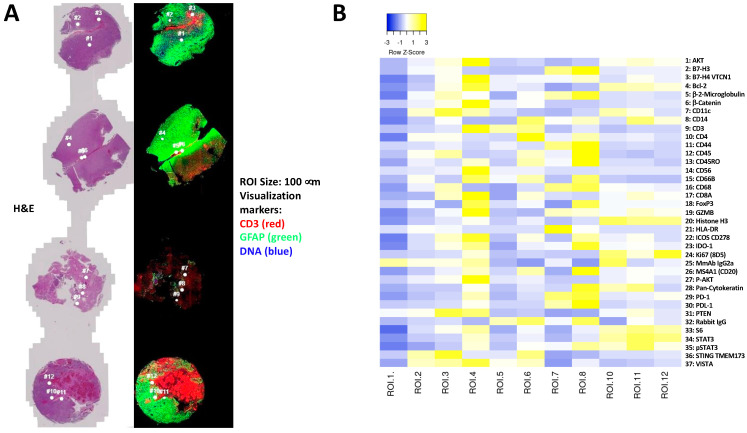
Immune partial proteomic profiling in primary and recurrent paired tumors using the GeoMx Digital Spatial Profiling (DSP) system. (**A**) Representative image acquired using the GeoMx DSP system of primary GBM from patient 1. Twelve regions of interest were analyzed (ROIs 1–12) according to morphological features. H&E staining: left; CD3, GFAP et DNA labeling right. Analyses were performed on primary and recurrent GBM from the five patients. (**B**) Heatmap depicting unsupervised clustering of the expression of the 39 antigens in the 12 ROIs in the primary GBM of patient 1. (**C**) ROIs from patients 1 to 5 were chosen to analyze the immune markers in the “center” versus the “periphery” of the tumor in primary and recurrent GBM. (**D**) Heatmap depicting clustering of the expression of the 39 antigens in the primary versus recurrent paired tumors and in the center versus periphery of the tumors of the five patients.

**Figure 4 cancers-15-03256-f004:**
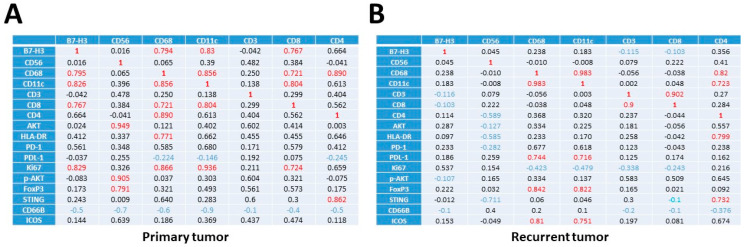
Correlation between immune marker expression. Matrix correlation between a selection of immune markers data from the five primary (**A**) and recurrent GBM (**B**). The X-axis and Y-axis indicate the antigens, and the Z-axis indicates the level of correlation between the two immune cells. Left primary and right recurrent GBM. (**C**) Expression of selected immune markers in ROIs: top lane checkpoint antigen, lane 2 immune cells markers, lane 3 tumor markers.

**Figure 5 cancers-15-03256-f005:**
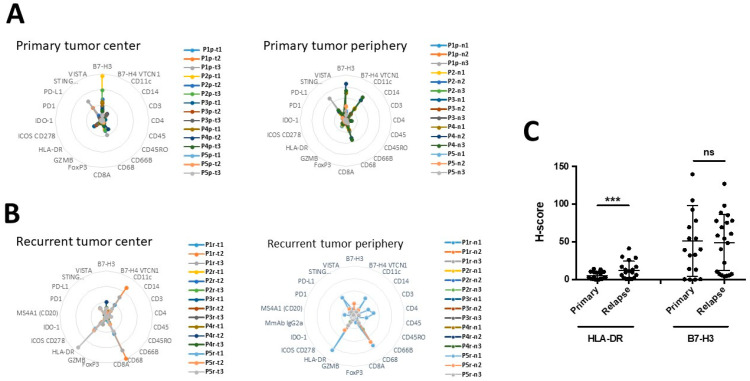
Analysis of the global expression of immune markers in the 15 primary and recurrent paired tumors. (**A**,**B**) Radar representation (Kiviat panel) from Geomx data shows the most prevalent markers of the immune system are in the center (**left**) and at the periphery (**right**) of primary (**A**) and recurrent (**B**) GBM. (**C**) H-Score of HLA-DR and B7-H3 in primary versus recurrent GBM. *** *p* < 0.001; ns = not significant.

**Table 1 cancers-15-03256-t001:** Characteristics of patients used in the study: sex; age at diagnosis; survival (mean and variability about the mean); brain localization; IDH/ATRX and p53 status.

Patients	Sex	Age	Period between Surgeries	Survival after 2nd Surgery	Overall Survival	Localization	Karnofsky (Diagnosis)	Karnofsky (Recurrent)	IDH		ATRX		P53	
									Primary	Recurrent	Primary	Recurrent	Primary	Recurrent
1	M	55	10.6	11	21.5	left temporal	90%	80%	wt	wt	+	+	+	−
2	F	56	8.6	8	16.6	left frontal	70%	50%	wt	wt	+	+	+	−
3	M	54	12.4	13	25.4	left frontal	90%	60%	wt	wt	+	partial loss	+	+
4	M	48	18.4	12	30.4	left frontal	90%	70%	wt	wt	+	+	+	−
5	M	42	15.7	16.5	26.7	right frontal	90%	60%	wt	wt	+	+	+	+
6	M	62	9	15.5	24.5	left temporal	70%	80%	wt	wt	+	+	−	+
7	F	51	26.8	25	51.8	right frontal	80%	80%	wt	wt	+	+	−	−
8	M	52	13.8	5	18.8	right frontal	50%	70%	wt	wt	+	+	+	−
9	M	43	16.2	9	25.2	left frontal	70%	80%	wt	wt	+	+	+	−
10	F	64	17.2	6	23.2	left temporal	70%	60%	wt	wt	nd	+	−	+
11	M	72	6.8	6.5	13.3	right frontal	80%	70%	wt	wt	+	partial loss	+	+
12	F	48	13.3	12	25.3	right parietal	50%	90%	wt	wt	+	+	+	+
13	M	72	12.6	19.2	31.8	right frontal	90%	90%	wt	wt	+	+	+	+
14	F	56	14.2	6.5	20.7	right parieto–occipital	80%	70%	wt	wt	+	+	+	−
15	M	49	12.7	11	23.7	left parietal	90%	80%	wt	wt	+	+	+	−

Distribution sex: F = 33%, M = 67%; mean age at diagnosis (years): 53.4 (F = 55; M = 52.6); mean overall survival (months): 24.85 ± 8.87.

## Data Availability

Data can be accessed upon request to the corresponding author: Francois M. Vallette (francois.vallette@univ-nantes.fr).

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
