# Peer review of "Spatial Distribution of Immune Cells in Primary and Recurrent Glioblastoma: A Small Case Study"

_cancers, 2023, doi:10.3390/cancers15123256_

Round 1
Reviewer 1 Report
The authors address a problem of great interest in the field of gliomas, making a longitudinal and spatial study of the heterogeneity of these tumors, especially at the level of the immune component. They use a small (but well-dissected) cohort of primary and recurrent tumor samples and a series of state-of-the-art techniques (nCounter and GeoMX analysis, for example) to analyze transcriptional profiles of different areas of the sampled tumors. In general, the idea of the manuscript is good although the cohort is a bit small to obtain big conclusions. Still, the results are interesting for the neurooncology field. However, the text needs some corrections and punctuations to be better understood.
Major points:
-Authors should explain better Figure 1D: Why the sum of the different immune cells is 100%? What is the contribution of the tumor cells?
-Authors should include a comparison between the IHC results and the nanostring or the Gmx data, is there a correlation? Please, discuss.
-Explanation of Figure 2 is confusing in the text. The conclusions are based on the clustering analysis? It says that primary and recurrent tumors from patients 3 and 5 are similar but in the Figure it looks like the strongest similarities are in patients 1 and 5. Authors should revise the explanation of that Figure in the text.
- In Figure 3, is there any correlation between the CD3 staining of the ROIs and the GeoMX quantification? If not, please explain.
- is there any correlation between the expression of HLA-DR or B7-H3, particularly in recurrent tumors, and survival of the 15 patients included in this study? Please explain or include that analysis.
- In the Discussion section, authors mention that there is an increase of CD68 and CD8 in recurrent tumors but Figure 1 only shows a change in CD68. Authors should revise that and correct or explain the sentence.
- Is it rabbit IgG a negative control? Should the samples be normalized with this marker? Please explain.
Minor points:
-Specify if the survival data from Table 1 is from the first or the second surgery. I understand it is from the second surgery (Survival delay) but is not clear in the text.
-Is Patient 5 still alive? how many months have passed since the surgery?
-Specify the location of the samples used in Figure 1 (core vs periphery)
-In the legend of Figure 3C it says that only Patient 1 is shown but there are images from all patients. Correct that or explain better.
There are some small typos and mistakes (capital letters missing or extra spaces) that should be revised throughout the text.
Author Response
Open Review 1
( ) I would not like to sign my review report
(x) I would like to sign my review report
Quality of English Language
( ) I am not qualified to assess the quality of English in this paper
( ) English very difficult to understand/incomprehensible
( ) Extensive editing of English language required
( ) Moderate editing of English language required
(x) Minor editing of English language required
( ) English language fine. No issues detected
|
Yes |
Can be improved |
Must be improved |
Not applicable |
|
|
Does the introduction provide sufficient background and include all relevant references? |
(x) |
( ) |
( ) |
( ) |
|
Are all the cited references relevant to the research? |
(x) |
( ) |
( ) |
( ) |
|
Is the research design appropriate? |
(x) |
( ) |
( ) |
( ) |
|
Are the methods adequately described? |
( ) |
(x) |
( ) |
( ) |
|
Are the results clearly presented? |
(x) |
( ) |
( ) |
( ) |
|
Are the conclusions supported by the results? |
(x) |
( ) |
( ) |
( ) |
Comments and Suggestions for Authors
The authors address a problem of great interest in the field of gliomas, making a longitudinal and spatial study of the heterogeneity of these tumors, especially at the level of the immune component. They use a small (but well-dissected) cohort of primary and recurrent tumor samples and a series of state-of-the-art techniques (nCounter and GeoMX analysis, for example) to analyze transcriptional profiles of different areas of the sampled tumors. In general, the idea of the manuscript is good although the cohort is a bit small to obtain big conclusions. Still, the results are interesting for the neurooncology field. However, the text needs some corrections and punctuations to be better understood.
We would like to thank the reviewer for these positive remarks. We are aware that our cohort is small (hence the “small case study” in the title) but re-operated GBM are scarce. However, we hope that the publication of this paper would be the cornerstone for a study with more patients.
Major points:
-Authors should explain better Figure 1D: Why the sum of the different immune cells is 100%? What is the contribution of the tumor cells?
Figure 1D indicates the proportion of CD3/CD8/CD68/PD-1/PDL-1 and the analysis has been made for the whole tumor. The proportion of one marker was compared to the others within each tumor. This has been specified in the legend of the figure.
-Authors should include a comparison between the IHC results and the nanostring or the Gmx data, is there a correlation? Please, discuss.
It is difficult to compare IHC and nanostring results as in one case it’s whole tumor analysis and in the other case a selection of ROIs. In addition, the Gmx is a quantification of antigens while IHC measures the percentage of positive cell whatever the level of expression of the antigen.
-Explanation of Figure 2 is confusing in the text. The conclusions are based on the clustering analysis? It says that primary and recurrent tumors from patients 3 and 5 are similar but in the Figure it looks like the strongest similarities are in patients 1 and 5. Authors should revise the explanation of that Figure in the text.
The analysis is based on unsupervised learning method. The reviewer is right patient 1 and 5 exhibit more similarities and this has been corrected in the text accordingly.
- In Figure 3, is there any correlation between the CD3 staining of the ROIs and the GeoMX quantification? If not, please explain.
The ROIs were indeed chosen among CD3+ areas.
- is there any correlation between the expression of HLA-DR or B7-H3, particularly in recurrent tumors, and survival of the 15 patients included in this study? Please explain or include that analysis.
The survival rate of patients with high content of HLA-DR was lower than those of patients with low content of HLA-DR (e.g Diao J, Xia T, Zhao H, Liu J, Li B, Zhang Z. Overexpression of HLA-DR is associated with prognosis of glioma patients. Int J Clin Exp Pathol. 2015 May 1;8(5):5485-90. PMID: 26191254; PMCID: PMC4503125). Similar results were obtained with B7H3, where it has been associated with tumor aggressiveness and poor prognosis (e.g Wang Z, Wang Z, Zhang C, Liu X, Li G, Liu S et al (2018) Genetic and clinical characterization of B7-H3 (CD276) expression and epigenetic regulation in diffuse brain glioma. Cancer Sci 109:2697–2705. https://doi.org/10.1111/cas.13744).
- In the Discussion section, authors mention that there is an increase of CD68 and CD8 in recurrent tumors but Figure 1 only shows a change in CD68. Authors should revise that and correct or explain the sentence.
The reviewer is right, we are sorry for the mistake. This has been corrected in the text.
- Is it rabbit IgG a negative control? Should the samples be normalized with this marker? Please explain.
Normalization includes rabbit IgG and was performed as described in the Nanostring protocol.
Minor points:
-Specify if the survival data from Table 1 is from the first or the second surgery. I understand it is from the second surgery (Survival delay) but is not clear in the text.
-Is Patient 5 still alive? how many months have passed since the surgery?
The table has been updated.
-Specify the location of the samples used in Figure 1 (core vs periphery)
The samples were taken on the whole tumor with no specific location (positive cells in the tumor samples).
-In the legend of Figure 3C it says that only Patient 1 is shown but there are images from all patients. Correct that or explain better.
This has been corrected.
Comments on the Quality of English Language
There are some small typos and mistakes (capital letters missing or extra spaces) that should be revised throughout the text
We have tried to correct all the mistakes.
Reviewer 2 Report
The manuscript by Loussouarn et al., evaluates the spatial distribution of immune cells in primary and recurrent glioblastomas (GBM). To this aim matched primary and recurrent GBM are evaluated by 3 different approaches: immunohistochemistry with 5 antibodies, Geomix digital transcriptomic with a panel of 770 genes related to cancer-associated pathway, and Geomix spatial profiling of 37 proteins. The second and third technique are evaluated with a targeted approach to the central and marginal areas of the tumor, while IHC is not.
The manuscript is interesting but deserve attention in the following points:
Methods needs to be revised in the part relative to the Nanostring technology with antibodies, since it’s not clear where it is described. Is it in the “Nanostring genomic array” section? Information regarding the antibodies is also missing in Methods.
Radar representation of Figure 5A and B is not clear. Authors should explain in more detail the meaining in the text. In addition, CD11c seems to be present also in the center of the primary tumor. What is H score of Figure 5C?
Results regarding the presence of macrophages, evaluated as CD68+ cells, are contradicting. In fact, data obtained with IHC indicate that “almost all tumors (both primary and recurrent) showed a high percentage of CD68+ cells”, lines 200-201, while data from Figure 5 (is it Geomix proteomic? It’s not indicated in the text of the result section, nor in the legend, please specify) indicate that “CD68 and CD11c were found only in the periphery of the (primary) tumor” (line 344). Although IHC was performed without discrimination between central and peripheral area, I find difficult to reconciliate these findings and to sustain that in primary GBM macrophages are not present in the central area, especially because several published manuscripts indicate, instead, a large presence of such cells. Please explain.
In figure 1C only CD68+ cells are increased significantly, but not CD8+ T cells. However, in Discussion Authors state that “cytotoxic T lymphocytes increased during tumor recurrence”. Please resolve this discrepancy.
Regarding the sentence “CD11c is a pro-tumoral macrophage marker” (lines 303-304), please add a reference.
Fine for me. Only a few typos are present.
Author Response
Review 2
Open Review
( ) I would not like to sign my review report
(x) I would like to sign my review report
Quality of English Language
( ) I am not qualified to assess the quality of English in this paper
( ) English very difficult to understand/incomprehensible
( ) Extensive editing of English language required
( ) Moderate editing of English language required
(x) Minor editing of English language required
( ) English language fine. No issues detected
|
Yes |
Can be improved |
Must be improved |
Not applicable |
|
|
Does the introduction provide sufficient background and include all relevant references? |
(x) |
( ) |
( ) |
( ) |
|
Are all the cited references relevant to the research? |
( ) |
(x) |
( ) |
( ) |
|
Is the research design appropriate? |
(x) |
( ) |
( ) |
( ) |
|
Are the methods adequately described? |
( ) |
( ) |
(x) |
( ) |
|
Are the results clearly presented? |
( ) |
(x) |
( ) |
( ) |
|
Are the conclusions supported by the results? |
(x) |
( ) |
( ) |
( ) |
Comments and Suggestions for Authors
The manuscript by Loussouarn et al., evaluates the spatial distribution of immune cells in primary and recurrent glioblastomas (GBM). To this aim matched primary and recurrent GBM are evaluated by 3 different approaches: immunohistochemistry with 5 antibodies, Geomix digital transcriptomic with a panel of 770 genes related to cancer-associated pathway, and Geomix spatial profiling of 37 proteins. The second and third technique are evaluated with a targeted approach to the central and marginal areas of the tumor, while IHC is not.
The manuscript is interesting but deserve attention in the following points:
Methods needs to be revised in the part relative to the Nanostring technology with antibodies, since it’s not clear where it is described. Is it in the “Nanostring genomic array” section? Information regarding the antibodies is also missing in Methods.
We have rearranged the methods section accordingly. However, the protocol can be find in , https://nanostring.com/wp-content/uploads/WP_GeoMx_Antibody_Validation_White_Paper.pdf.
Radar representation of Figure 5A and B is not clear. Authors should explain in more detail the meaining in the text. In addition, CD11c seems to be present also in the center of the primary tumor. What is H score of Figure 5C?
A radar chart is a way of showing multiple data points and the variation between them in the center and the periphery of the tumor determined by the position of the ROIs. Figure 5 A and B indicates that the main antigens expressed in the primary and recurrent tumors. A radar chart is a way of showing multiple data points and the variation between them the reviewer is right, our results indicate a change in CD11C localization from primary to the recurrent tumor. This has been added to the text: “It is noteworthy that CD11c localization appears to be different in primary versus GBM as it I predominantly present in the periphery of the primary tumors versus the center of recurrent tumors”.
The H-score for B7H3 and HLADR is indicated in the X-axis.
Results regarding the presence of macrophages, evaluated as CD68+ cells, are contradicting. In fact, data obtained with IHC indicate that “almost all tumors (both primary and recurrent) showed a high percentage of CD68+ cells”, lines 200-201, while data from Figure 5 (is it Geomix proteomic? It’s not indicated in the text of the result section, nor in the legend, please specify) indicate that “CD68 and CD11c were found only in the periphery of the (primary) tumor” (line 344). Although IHC was performed without discrimination between central and peripheral area, I find difficult to reconciliate these findings and to sustain that in primary GBM macrophages are not present in the central area, especially because several published manuscripts indicate, instead, a large presence of such cells. Please explain.
Our results indicate that CD11c and CD68 can be differently localized in primary and recurrent GBM but still macrophages are present in both compartments albeit in different localization.
Qualitative vs quantitative
In figure 1C only CD68+ cells are increased significantly, but not CD8+ T cells. However, in Discussion Authors state that “cytotoxic T lymphocytes increased during tumor recurrence”. Please resolve this discrepancy.
This has been corrected in the text.
Regarding the sentence “CD11c is a pro-tumoral macrophage marker” (lines 303-304), please add a reference.
This reference has been added to the text : Kokubu, Y., Tabu, K., Muramatsu, N., Wang, W., Murota, Y., Nobuhisa, I., Jinushi, M. and Taga, T. (2016), Induction of protumoral CD11c high macrophages by glioma cancer stem cells through GM-CSF. Genes Cells, 21: 241-251. https://doi.org/10.1111/gtc.12333
Comments on the Quality of English Language
Fine for me. Only a few typos are present.
Submission Date
15 May 2023
Date of this review
19 May 2023 15:47:01
Round 2
Reviewer 1 Report
I a happy with the author´s responses